# List-Decodable Mean Estimation via Iterative Multi-Filtering

**Ilias Diakonikolas**
Department of Computer Sciences
University of Wisconsin, Madison
Madison, WI 53706
ilias@cs.wisc.edu

**Daniel M. Kane**
Department of Computer Science
University of California, San Diego
La Jolla, CA 92093
dakane@cs.ucsd.edu

**Daniel Kongsgaard**
Department of Mathematics
University of California, San Diego
La Jolla, CA 92093
dkongsga@ucsd.edu

## Abstract

We study the problem of *list-decodable mean estimation* for bounded covariance distributions. Specifically, we are given a set $T$ of points in $\mathbb{R}^d$ with the promise that an unknown $\alpha$-fraction of points in $T$, where $0 < \alpha < 1/2$, are drawn from an unknown mean and bounded covariance distribution $D$, and no assumptions are made on the remaining points. The goal is to output a small list of hypothesis vectors such that at least one of them is close to the mean of $D$. We give the first practically viable estimator for this problem. In more detail, our algorithm is sample and computationally efficient, and achieves information-theoretically near-optimal error. While the only prior algorithm for this setting inherently relied on the ellipsoid method, our algorithm is iterative and only uses spectral techniques. Our main technical innovation is the design of a soft outlier removal procedure for high-dimensional heavy-tailed datasets with a majority of outliers.

## 1   Introduction

### 1.1   Background and Motivation

Estimating the mean of a high-dimensional distribution is one of the most fundamental statistical tasks. The standard assumption is that the input data are independent samples drawn from a known family of distributions. However, this is rarely true in practice and it is important to design estimators that are *robust* in the presence of outliers. In recent years, the design of outlier robust estimators has become a pressing challenge in several data analysis tasks, including in designing defenses against data poisoning [BNJT10, BNL12] and in analyzing biological datasets where natural outliers are common [RPW+02, PLJD10, LAT+08].

The field of robust statistics [HRRS86, HR09] traditionally studies the setting where the fraction of outliers is a small constant (smaller than $1/2$), and therefore the clean data is the majority of the input dataset. Classical work in this field pinned down the minimax risk of high-dimensional robust estimation in several settings of interest. In contrast, until relatively recently, our understanding of even the most basic computational questions was startlingly poor. Recent work in computer science, starting with [DKK+16, LRV16], gave the first efficient robust estimators for various high-dimensional statistical tasks, including mean estimation. Since the dissemination of [DKK+16,

LRV16], there has been significant research activity on designing efficient robust estimators in a variety of settings (see, e.g., [DKK$^+$17, BDLS17, DKS16, DKK$^+$18, CDKS18, KKM18, KS17, HL18, DKS19, DKK$^+$19a, DKK$^+$19b]). The reader is referred to [DK19] for a recent survey of the extensive recent literature.

The aforementioned literature studies the setting where the clean data (inliers) are the majority of the input dataset. *In this paper, we study the algorithmic problem of high-dimensional mean estimation in the more challenging regime where the fraction $\alpha$ of inliers is small – strictly smaller than $1/2$.* This regime is fundamental in its own right and is motivated by a number of machine learning applications, e.g., in crowdsourcing [SVC16, SKL17, MV18]).

Mean estimation with a majority of outliers was first studied in [CSV17]. We note that, in the $\alpha < 1/2$ regime, it is information-theoretically impossible to estimate the mean with a single hypothesis. Indeed, an adversary can produce $\Omega(1/\alpha)$ clusters of points each drawn from a "good" distribution with different mean. Even if the algorithm could learn the distribution of the samples exactly, it would still not be able to identify which cluster is the correct one. Hence, the definition of "learning" must be relaxed. In particular, the algorithm should be allowed to return *a small list of hypotheses* with the guarantee that *at least one* of the hypotheses is close to the true mean. This is the model of *list-decodable learning* [BBV08]. It should be noted that in list-decodable learning, it is often information-theoretically necessary for the error to increase as the fraction $\alpha$ goes to $0$.

[CSV17] gave an algorithm for list-decodable mean estimation on $\mathbb{R}^d$ under the assumption that the inliers are drawn from a distribution $D$ with bounded covariance, i.e., $\Sigma \preceq \sigma^2 I$. The [CSV17] algorithm has sample complexity $n = \Omega(d/\alpha)$, runs in $\mathrm{poly}(n, d, 1/\alpha)$ time, and outputs a list of $O(1/\alpha)$ hypotheses one of which is within $\ell_2$-distance $\tilde{O}(\alpha^{-1/2})$ from the true mean of $D$. The sample complexity of the aforementioned algorithm is optimal, within constant factors, and subsequent work [DKS18] showed that the information-theoretically optimal error is $\Theta(1/\alpha^{1/2})$ (upper and lower bound). Importantly, the [CSV17] algorithm relies on the ellipsoid method for convex programming. Consequently, its computational complexity, though polynomially bounded, is impractically high.

*The main motivation for the current paper is to design a fast, practically viable, algorithm for list-decodable mean estimation under minimal assumptions.* In the presence of a *minority* of outliers (i.e., for $\alpha > 1/2$), the iterative filtering method of [DKK$^+$16, DKK$^+$17] is a fast and practical algorithm which attains the information-theoretically optimal error under only a bounded covariance assumption. More recent work has also obtained near-linear time algorithms in this setting [CDG18a, DL19, DHL19]. In the list-decodable setting, however, progress on faster algorithms has been slower. Prior to the current work, the ellipsoid-based method of [CSV17] was the only known polynomial-time algorithm for mean estimation under a bounded covariance assumption. We note that a number of more recent works developed list-decoding algorithms for mean estimation, linear regression, and subspace recovery using the SoS convex programming hierarchy [KS17, KKK19, RY20a, BK20, RY20b]. In a departure from these convex optimization methods, [DKS18] obtained an iterative *spectral* list-decodable mean algorithm under the much stronger assumption that the good data is drawn from an identity covariance Gaussian. At a high-level, in this work we provide a broad generalization of the [DKS18] algorithm and techniques to all bounded covariance distributions.

## 1.2   Our Contributions

We start by defining the problem we study.

**Definition 1.1 (List-Decodable Mean Estimation.).**  Given a set $T$ of $n$ points in $\mathbb{R}^d$ and a parameter $\alpha \in (0, 1/2)$ such that an $\alpha$-fraction of the points in $T$ are i.i.d. samples from a distribution $D$ with unknown mean $\mu$ and unknown covariance $\Sigma \preceq \sigma^2 I$, we want to output a list of $s = \mathrm{poly}(1/\alpha)$ candidate vectors $\{\widehat{\mu}_i\}_{i \in [s]}$ such that with high probability we have that $\min_{i \in [s]} \|\widehat{\mu}_i - \mu\|_2$ is small.

Some comments are in order: First, we emphasize that no assumptions are made on the remaining $(1 - \alpha)$-fraction of the points in $T$. These points can be arbitrary and may be chosen by an adversary that is computationally unbounded and is allowed to inspect the set of inliers. The information-theoretically best possible size of the hypotheses list is $s = \Theta(1/\alpha)$. Moreover, if we are given a list of $s = \mathrm{poly}(1/\alpha)$ hypotheses one of which is accurate, we can efficiently post-process them to obtain an $O(1/\alpha)$-sized list with nearly the same error guarantee, see, e.g., Proposition B.1 of

[DKS18] and Corollary 2.16 of [Ste18]. For completeness, in Appendix C, we provide a simple and self-contained method.

In this work, we give an iterative spectral algorithm for list-decodable mean estimation under only a bounded covariance assumption that matches the sample complexity and accuracy of the previous ellipsoid-based algorithm [CSV17] while being significantly faster and potentially practical.

**Theorem 1.2 (Main Algorithmic Result).** *Let $T$ be a set of $n = \Omega(d/\alpha)$ points in $\mathbb{R}^d$ with the promise that an unknown $\alpha$-fraction of points in $T$, $0 < \alpha < 1/2$, are drawn from a distribution $D$ with unknown bounded covariance $\Sigma \preceq \sigma^2 I$. There is an algorithm that, on input $T$ and $\alpha$, runs in $\tilde{O}(n^2 d/\alpha^2)$ time and outputs a list of $O(1/\alpha^2)$ hypothesis vectors such that with high probability at least one of these vectors is within $\ell_2$-distance $O(\sigma \log(1/\alpha)/\sqrt{\alpha})$ from the mean of $D$.*

**Discussion** Before we proceed, we provide a few remarks about the performance of our new algorithm establishing Theorem 1.2. First, we note that the sample complexity of our algorithm is $O(d/\alpha)$, which is optimal within constant factors, and its error guarantee is $O(\sigma \log(1/\alpha)/\sqrt{\alpha})$, which is optimal up to the $O(\log(1/\alpha))$ factor. We now comment on the running time. Our algorithm is iterative with every iteration running in near-linear time $\tilde{O}(nd)$. The dominant operation in a given iteration is the computation of an approximately largest eigenvector/eigenvalue of an empirical covariance matrix, which can be implemented in $\tilde{O}(nd)$ time by power iteration. The overall running time follows from a worst-case upper bound of $O(n)$ on the total number of iterations. We expect that the number of iterations will be much smaller for reasonable instances, as has been observed experimentally for analogous iterative algorithms for the large $\alpha$ case [DKK$^+$17, DKK$^+$19a]. Finally, as we show in Appendix C, there is a simple and efficient post-processing algorithm that outputs a list of size $O(1/\alpha)$ without affecting the runtime or error guarantee by more than a constant factor.

**Application to Learning Mixture Models** As observed in [CSV17], list-decoding generalizes the problem of learning mixtures. Specifically, a list-decodable mean algorithm for bounded covariance distributions can be used in a black-box manner (by treating a single cluster as the set of inliers) to obtain an accurate clustering for mixtures of bounded covariance distributions. If each distribution in the mixture has unknown covariance bounded by $\sigma^2 I$, and the means of the components are separated by $\tilde{\Omega}(\sigma/\sqrt{\alpha})$, we can perform accurate clustering, even in the presence of a small fraction of adversarial outliers. This implication was shown in [CSV17]. Our new algorithm for list-decodable mean estimation gives a simpler and faster method for this problem.

**Technical Overview** Here we describe our techniques in tandem with a comparison to prior work.

The "filtering" framework [DKK$^+$16, DKK$^+$17] works by iteratively detecting and removing outliers until the empirical variance in every direction is not much larger than expected. If every direction has small empirical variance, this certifies that the the empirical mean is close to target mean. Otherwise, a filtering algorithm projects the points in a direction of large variance and removes (or reduces the weight of) those points whose projections lie unexpectedly far from the empirical median in this direction. In the small $\alpha$ setting, the one-dimensional "outlier removal" procedure is necessarily more complicated. For example, the input distribution can simulate a mixture of $1/\alpha$ many Gaussians whose means are far from each other, and the algorithm will have no way of knowing which is the real one. To address this issue, one requires a more elaborate method, which we call a *multifilter*. A multifilter can return several (potentially overlapping) subsets of the original dataset with the guarantee that *at least one* of these subsets is substantially "cleaner". This idea was introduced in [DKS18], who gave a multifilter for identity covariance Gaussians with error $\tilde{O}(\alpha^{-1/2})$. The multifilter of [DKS18] makes essential use of the fact that the covariance of the inliers is known and that the Gaussian distribution has very strong concentration. In this work, we build on [DKS18] to develop a multifilter for bounded covariance distributions.

We start by describing the Gaussian multifilter [DKS18]. Suppose we have found a large variance direction. After we project the data in such a direction, there are two cases to consider. The first is when almost all of the samples lie in some relatively short interval $I$. In this case, the target mean must lie in that interval (as otherwise an approximately $\alpha/2$ fraction of the good samples must lie outside of this interval), and then samples that lie too far from this interval $I$ are almost certainly outliers. The other case is more complicated. If $\alpha < 1/2$, there might be multiple clusters of points which contain an $\alpha$ fraction of the samples and could reasonably contain the inliers. If some pair of these clusters lie far from each other, we might not be able to reduce the variance in this direction

simply by removing obvious outliers. In this case, [DKS18] find a pair of overlapping intervals $I_1$ and $I_2$ such that with high probability either almost all the inliers lie in $I_1$ or almost all the inliers lie in $I_2$. The algorithm then recurses on both $I_1$ and $I_2$. To ensure that the complexity of the algorithm does not blow-up with the recursion, [DKS18] require that the sum of the squares of the numbers of remaining samples in each subinterval is at most the square of the original number of samples.

At a high-level, our algorithm follows the same framework. However, there were several key places where [DKS18] used the strong concentration bounds of the Gaussian assumption that we cannot use in our context. For example, in the case where most of the samples are contained within an interval $I$, Gaussian concentration bounds imply that almost all of the good samples lie within distance $O(\sqrt{\log(1/\alpha)})$ of the interval $I$, and therefore that almost all samples outside of this range will be outliers. This is of course not true for heavy-tailed data. To address this issue, we employ a soft-outlier procedure that reduces the weight of each point based on its squared distance from $I$. The analysis in this case is much more subtle than in the Gaussian setting.

The other more serious issue comes from the multi-filter case. With Gaussian tails, so long as the subintervals $I_1$ and $I_2$ overlap for a distance of $O(\sqrt{\log(1/\alpha)})$, this suffices to guarantee that the correct choice of interval only throws away a $\text{poly}(\alpha)$-fraction of the good points. As long as at least an $\alpha$-fraction of the total points are being removed, it is easy to see that this is sufficient. From there it is relatively easy to show that, unless almost all of the points are contained in some small interval, some appropriate subintervals $I_1$ and $I_2$ can be found. For bounded covariance distributions, our generalization of this case is more complicated. In order to ensure that the fraction of good samples lost is small, even if the true mean is exactly in the middle of the overlap between $I_1$ and $I_2$, we might need to make this overlap quite large. In particular, in contrast to the Gaussian case, we cannot afford to ensure that some small $\text{poly}(\alpha)$ fraction of the inliers are lost. In fact, we will need to adapt the fraction of inliers we are willing to lose to the number of total points lost and ensure that the fraction of inliers removed is *substantially better* than the fraction of outliers removed (namely, by a $\log(1/\alpha)$ factor). This step is necessary for our new analysis of the behavior of the algorithm under repeated applications of the multifilter. With this careful tuning, we can show that there will be an appropriate pair of intervals, unless the distribution of points along the critical direction satisfy inverse-quadratic tail bounds. This is not enough to show that there is a short interval $I$ containing almost all of the points, but it will turn out to be enough to show the existence of an $I$ containing almost all of the points for which the variance of the points within $I$ is not too large. This turns out to be sufficient for our analysis of the other case.

**Concurrent and Independent Work** Contemporaneous work [CMY20], using different techniques, gave an algorithm for the same problem with asymptotic running time $\tilde{O}(nd/\alpha^c)$, for some (unspecified) constant $c$. At a high-level, the algorithm of [CMY20] builds on the convex optimization frameworks of [DKK+16, CDG18b], leveraging faster algorithms for solving structured SDPs.

## 2 Preliminaries

**Notation** We write $\lg = \log_2$. For an interval $I = [a, b] = [t - R, t + R]$, we will write $2I = [t - 2R, t + 2R]$. For a vector $v$, $\|v\|_2$ denotes its Euclidean norm. For a symmetric matrix $M$, $\|M\|_2$ denotes its spectral norm. We will use $\preceq$ to denote the Loewner ordering between matrices, i.e., for symmetric matrices $A, B$, we will write $A \preceq B$ to denote that $B - A$ is positive semidefinite.

For a set $T \subset \mathbb{R}^d$ we will often attach a weight function $w \colon T \to [0, 1]$ and write $w(R) = \sum_{x \in R} w(x)$ for any subset $R \subseteq T$. We will furthermore denote weighted mean, weighted covariance matrix, and weighted variance (in a given direction $v$) with respect to the weight function $w$ by $\mu_w(R) = \mathbb{E}_w[R] = \frac{1}{w(R)} \sum_{x \in R} w(x)x$, $\text{Cov}_w[R] = \frac{1}{w(R)} \sum_{x \in R} w(x)(x - \mu_w(R))(x - \mu_w(R))^T$, and $\text{Var}_w[v \cdot R] = \frac{1}{w(R)} \sum_{x \in R} w(x)(v \cdot x - v \cdot \mu_w(R))^2$ for a subset $R \subseteq T$. When the underlying weight function $w$ assigns the same weight on each point, we will drop the index $w$ from these quantities. For example, we will use $\mu(R)$ and $\text{Cov}[R]$ for the empirical mean and covariance under the uniform distribution on the set $R$. Furthermore, we will write $w\text{-Pr}$ for the weighted probability with respect to the weight function $w$ and $\text{Pr}$ for the usual (counting) probability on sets.

# 3 Algorithm and Analysis

In Section 3.1, we give a deterministic condition under which our algorithm succeeds and bound the number of samples needed to guarantee that this condition holds with high probability. In Section 3.2, we present our basic multifilter. In Section 3.3, we show how to use the basic multifilter to obtain our list-decoding learning algorithm. We conclude with some open problems in Section 4.

Due to space limitations, some proofs are deferred to Appendix A. In Appendix B, we analyze the running time of our algorithm. Finally, in Appendix C, we show how to efficiently post-process the output of our main algorithm.

## 3.1 Setup and Main Theorem

We define the following deterministic condition on the set of clean samples.

**Definition 3.1 (Representative set).** Let $D$ be a distribution on $\mathbb{R}^d$ with mean $\mu$ and covariance $\Sigma \preceq I$. A set $S \subset \mathbb{R}^d$ is *representative* (with respect to $D$) if $\|\mathrm{Cov}[S]\|_2 \leq 1$ and $\|\mu(S) - \mu\|_2 \leq 1$.

Our algorithm requires the following notion of goodness for the corrupted set $T$.

**Definition 3.2 (Good set).** Let $D$ be a distribution on $\mathbb{R}^d$ with mean $\mu$ and covariance $\Sigma \preceq I$, and let $0 < \alpha < 1/2$. A set $T \subset \mathbb{R}^d$ is said to be $\alpha$-good (with respect to $D$) if there exists $S \subseteq T$ which is representative (with respect to $D$) and satisfies $|S| \geq \alpha |T|$.

In Sections 3.2 and 3.3, we prove the following theorem:

**Theorem 3.3 (Main Theorem).** *Suppose that $T$ is $\alpha$-good with respect to a distribution $D$ on $\mathbb{R}^d$. Then the algorithm* LIST-DECODE-MEAN *runs in time $\tilde{O}(|T|^2 d/\alpha^2)$ and outputs a list of $O(1/\alpha^2)$ hypothesis vectors at least one of which has $\ell_2$-distance $O(\log(1/\alpha)/\sqrt{\alpha})$ from the mean of $D$.*

**Sample Complexity.** The deterministic conditions of Definitions 3.1 and 3.2 hold with high probability if the set $T$ has size $n = |T| = \Omega(d/\alpha)$. Note that $T$ contains a subset $G$ of $\alpha n \geq d$ i.i.d. samples from the distribution $D$. The following lemma shows that with high probability $G$ contains a subset $S$ such that $|S| \geq |G|/2$ that satisfies the properties of a representative set, up to rescaling.

**Lemma 3.4 (see, e.g., Proposition 1.1 in [CSV17]).** *Let $D$ be a distribution on $\mathbb{R}^d$ with covariance matrix $\Sigma \preceq \sigma^2 I$, $\sigma > 0$, and $G$ be a multiset of $n \geq d$ i.i.d. samples from $D$. Then, with high probability, there exists a subset $S \subseteq G$ of size $|S| \geq |G|/2$ such that $\|\mathrm{Cov}[S]\|_2 \leq c\,\sigma^2$ and $\|\mu(S) - \mu\|_2 \leq c\,\sigma$, where $c > 1$ is a universal constant independent of $D$.*

We henceforth condition on the conclusions of Lemma 3.4 holding. Note that by dividing each of our samples by $c\,\sigma$ we obtain a representative set $S$ with respect to the distribution $(1/(c\sigma))D$. Also note that the corrupted set $T$ will be $\alpha/2$-good. By Theorem 3.3, we thus obtain a list of hypothesis one of which has $\ell_2$-error $O(\log(1/\alpha)/\sqrt{\alpha})$. By rescaling back, we get an estimate of the true mean $\mu$ of $D$ within $\ell_2$ error $O(\sigma \log(1/\alpha)/\sqrt{\alpha})$, as desired. This proves Theorem 1.2.

Throughout this section, we will denote by $T$ the initial corrupted set of points and by $S \subset T$ a representative set with $|S| \geq \alpha |T|$.

## 3.2 Basic Multifilter

The basic multifilter is a key subroutine of our algorithm. Intuitively, it takes as input a large variance direction and, under certain assumptions, splits the dataset into at most two (overlapping) datasets at least one of which is cleaner. Since we are employing a soft outlier removal procedure, the real version of the routine starts from a weight function on the dataset $T$ and produces one or two weight functions on $T$ with desirable properties.

In the body of this subsection, we show that the BASICMULTIFILTER algorithm has certain desirable properties that we will later use to establish correctness of our main algorithm.

The following notation will facilitate our analysis. We will denote $\Delta w(S) = w(S) - w_{\mathrm{new}}(S)$ to describe the change of weights during a step of the BASICMULTIFILTER algorithm.

---

**Algorithm** BASICMULTIFILTER

Input: unit vector $v \in \mathbb{R}^d$, $T \subset \mathbb{R}^d$ and weight function $w$ on $T$, $0 < \alpha < 1/2$

1. Let $C > 0$ be a sufficiently large universal constant.

2. Let $a \in \mathbb{R}$ be maximal such that $w(\{x \in T : v \cdot x < a\}) \leq \alpha w(T)/8$ and $b$ be minimal such that $w(\{x \in T : v \cdot x > b\}) \leq \alpha w(T)/8$. Let $I = [a, b]$.

3. If $\mathrm{Var}_w[v \cdot T \cap 2I] \leq C \cdot \log(2/\alpha)^2$, then

   (a) If $\mathrm{Var}_w[v \cdot T] \leq 2C \cdot \log(2/\alpha)^2$, return "YES".

   (b) Let $f(x) = \min_{t \in [a,b]} |v \cdot x - t|^2$, and redefine the weight of each $x \in T$ by

   $$w_{\mathrm{new}}(x) = \Big(1 - \frac{f(x)}{\max_{x \in T} f(x)}\Big) w(x).$$

   (c) Return $\{(T, w_{\mathrm{new}}, \alpha)\}$.

4. If $I$ does not satisfy the condition of Step 3., then

   (a) Find $t \in \mathbb{R}$ and $R > 0$ such that the sets $T_1 = \{x \in T : v \cdot x \geq t - R\}$ and $T_2 = \{x \in T : v \cdot x < t + R\}$ satisfy

   $$w(T_1)^2 + w(T_2)^2 \leq w(T)^2 \ , \tag{1}$$

   and

   $$\min\Big(1 - \frac{w(T_1)}{w(T)}, 1 - \frac{w(T_2)}{w(T)}\Big) \geq \frac{48 \lg(2/\alpha)}{R^2} \ . \tag{2}$$

   Define two weight functions $w^{(1)}$ and $w^{(2)}$ on $T$ by multiplying the indicator functions of $T_1$ and $T_2$ with the weight function $w$.

   (b) Return $\{(T, w^{(1)}, \alpha), (T, w^{(2)}, \alpha)\}$.

---

Our first lemma bounds the relative change in the weight of $S$ and $T$ if the BASICMULTIFILTER algorithm outputs a single weight function $w_{\mathrm{new}}$ in Step 3.(c).

**Lemma 3.5.** *If $T$ is $\alpha$-good and $w(S) \geq 3|S|/4$, then after Step 3.(b) of* BASICMULTIFILTER *we have $\frac{\Delta w(S)}{w(S)} \leq \frac{\Delta w(T)}{w(T)} \cdot \frac{1}{24 \lg(2/\alpha)}$.*

*Proof (sketch).* We give a proof sketch without the constant of 24. For the full proof, see Appendix A.

Firstly, we note that $v \cdot \mu_w(S) \in [a - O(1), b + O(1)]$. This is because if, say $\mu_w(S)$ was much less than $a$, then since all but a small fraction of the points in $S$ have $v \cdot (x - \mu_w(S)) = O(1)$, this would imply that most of the points of $S$ are less than $a$. But since all but a $1/4$-fraction of the points of $S$ remain under weight $w$ and since they account for at least an $\alpha$ fraction of the weight of $T$, this would imply that more than an $\alpha/8$-fraction of the weight of $T$ was less than $a$, which is a contradiction.

Given this, we have that $f(x) = O(1 + (v \cdot (x - \mu_w(S)))^2)$ and therefore the average value of $f$ over $S$ is $O(1)$. On the other hand,

$$\mathrm{Var}_w[v \cdot T] \leq \mathrm{Var}_w[v \cdot T \cap 2I] + O(\mathbb{E}_w[f(T)]) \ .$$

This implies that since $\mathrm{Var}_w[v \cdot T]$ is large, $\mathbb{E}_w[f(T)]$ is $\Omega(\log^2(1/\alpha))$.

Finally, since we are downweighting point $x$ by an amount proportional to $f(x)$, it is easy to see that $\Delta w(T)/w(T)$ is proportional to $\mathbb{E}_w[f(T)]$, while $\Delta w(S)/w(S)$ is proportional to $\mathbb{E}_w[f(S)]$, and the lemma follows. $\qquad\square$

Our second lemma says that conditions (1) and (2) in Step 4.(a) of the algorithm are satisfiable.

**Lemma 3.6.** *If* BASICMULTIFILTER *reaches Step 4.(a), there exist $t \in \mathbb{R}$ and $R > 0$ such that the conditions* (1) *and* (2) *are satisfied.*

*Proof.* For $t \in \mathbb{R}$ and $R > 0$, we will use the notation $g(t+R) = 1 - \frac{w(T_2)}{w(T)}$ and $g^c(t-R) = 1 - \frac{w(T_1)}{w(T)}$ to describe the tails of the weight distribution. Thus, (1) and (2) become

$$(1 - g^c(t - R))^2 + (1 - g(t + R))^2 \leq 1 \tag{3}$$

and

$$\min(g^c(t - R), g(t + R)) \geq 48 \lg(2/\alpha)/R^2 \ . \tag{4}$$

Now assume for contradiction that we cannot find any $t \in \mathbb{R}$ and $R > 0$ satisfying both (3) and (4), i.e., either (3) fails or (4) fails. Let $\text{med} = \text{median}_w(v \cdot T)$.

Let $x = x_0 > \text{med}$ and let $\gamma = \gamma_0 = g(x_0)$, and note that $\gamma_0 \leq 1/2$. We want to show that

$$x \leq \text{med} + O\left(\sqrt{\lg(2/\alpha)/\gamma}\right). \tag{5}$$

First find $t_0$ and $R_0$ such that $x_0 = t_0 + R_0$ and $\gamma_0 = 48 \lg(2/\alpha)/R_0^2$, i.e., $R_0 = \sqrt{48 \lg(2/\alpha)/\gamma_0}$. Then either $t_0 - R_0 \leq \text{med}$ or $t_0 - R_0 > \text{med}$. If $t_0 - R_0 \leq \text{med}$, then $x = t_0 + R_0 \leq \text{med} + 2R_0$ and we indeed get (5). On the other hand, if $t_0 - R_0 > \text{med}$ we see that $g^c(t_0 - R_0) \geq 1/2 \geq \gamma_0$, so (4) is satisfied. Thus, (3) must fail (by assumption), i.e., $g(t_0 - R_0)^2 + (1 - \gamma_0)^2 > 1$, since $g(t_0 - R_0) = 1 - g^c(t_0 - R_0)$. So

$$g(t_0 - R_0)^2 > 1 - (1 - \gamma_0)^2 = 2\gamma_0 - \gamma_0^2 = \gamma_0 + (\gamma_0 - \gamma_0^2) > \gamma_0,$$

and thus

$$g(x - 2R_0) = g(x_0 - 2R_0) = g(t_0 - R_0) > \sqrt{\gamma_0} = \gamma^{1/2}.$$

Now let $x_1 = x_0 - 2R_0 > \text{med}$ and let $\gamma_1 = g(x_1) \leq 1/2$. Note that $\gamma_1 > \sqrt{\gamma_0}$. By finding $t_1$ and $R_1$ as before and following the same argument, we get that

$$g(x - 2R_0 - 2R_1) = g(x_1 - 2R_1) = g(t_1 - R_1) > \sqrt{\gamma_1} > \gamma^{1/2^2}.$$

Continuing like this, we inductively get that

$$g(x_n) = g\left(x - 2\sum_{i=0}^{n-1} R_i\right) > \sqrt{\gamma_{n-1}} > \gamma^{1/2^n},$$

as long as $x_{n-1} = x - 2\sum_{i=0}^{n-2} R_i > \text{med}$. Hence $\gamma_n = g(x_n) > \gamma^{1/2^n}$, and thus $x_{\lg\lg(1/\gamma)} < \text{med}$ since $g(x_{\lg\lg(1/\gamma)}) > 1/2$. Therefore

$$\text{med} > x_{\lg\lg(1/\gamma)} = x - 2\sum_{i=0}^{\lg\lg(1/\gamma)-1} R_i = x - 2\sqrt{48\lg(2/\alpha)} \sum_{i=0}^{\lg\lg(1/\gamma)-1} \frac{1}{\sqrt{\gamma_i}}$$

$$\geq x - O\left(\sqrt{\lg(1/\alpha)} \sum_{i=1}^{\lg\lg(1/\gamma)} \frac{1}{\gamma^{1/2^i}}\right) \geq x - O\left(\frac{\sqrt{\lg(2/\alpha)}}{\sqrt{\gamma}}\right),$$

i.e., $x \leq \text{med} + O\left(\sqrt{\lg(2/\alpha)/\gamma}\right)$. Now writing $\gamma = g(\text{med} + t)$, for $t > 0$, the above gives that $t \leq O\left(\sqrt{\lg(2/\alpha)/\gamma}\right)$, and thus

$$\underset{y \in T}{w\text{-Pr}}[v \cdot y > \text{med} + t] = O(g(\text{med} + t)) = O(\gamma) \leq O\left(\lg(2/\alpha)/t^2\right).$$

A very similar proof yields the analogous result for $g^c(m - t)$, so $w\text{-Pr}_{y \in T}[|v \cdot y - \text{med}| > t] \leq O\left(\lg(2/\alpha)/t^2\right)$. Letting $a$ and $b$ be as in Step 2. of BASICMULTIFILTER, we note that

$$g(b - 1) = w\big(\{x \in T : v \cdot x \geq b - 1\}\big)/w(T) \geq \alpha/8$$

by the definition of $b$, so

$$b - 1 \leq \text{med} + O\left(\sqrt{\lg(2/\alpha)}/\sqrt{\alpha/4}\right) \leq \text{med} + O(1/\alpha),$$

and thus $b \leq \text{med} + O(1/\alpha)$. An analogous argument yields a similar result for $g^c(a)$, so $2I \subset [\text{med} - O(1/\alpha), \text{med} + O(1/\alpha)]$.

Finally we note that $w(\{y \in T : v \cdot y \notin 2I\}) \leq \alpha w(T)/4 \leq w(T)/2$, so

$$w\big(\{y \in T : v \cdot y \in 2I\}\big) = w(T) - w\big(\{y \in T : v \cdot y \notin 2I\}\big) \geq w(T) - w(T)/2 = w(T)/2,$$

and thus

$$\underset{y \in \{z \in T : v \cdot z \in 2I\}}{w\text{-Pr}}[|v \cdot y - \text{med}| > t] = \frac{w\big(\{y \in T : |v \cdot y - \text{med}| > t \text{ and } v \cdot y \in 2I\}\big)}{w\big(\{y \in T : v \cdot y \in 2I\}\big)}$$

$$\leq \frac{w\big(\{y \in T : |v \cdot y - \text{med}| > t\}\big)}{w(T)/2}$$

$$= 2\, \underset{y \in T}{w\text{-Pr}}[|v \cdot y - \text{med}| > t].$$

Hence, we have that

$$\text{Var}_w[v \cdot T \cap 2I] \le 2 \int_0^{O(1/\alpha)} 2t \cdot \underset{y \in T}{w\text{-Pr}}[|v \cdot y - \text{med}| > t]dt \le O(\lg(2/\alpha)) \int_1^{O(1/\alpha)} (1/t)dt$$
$$= O(\log(2/\alpha)^2).$$

Thus, if conditions (1) and (2) were not satisfiable, the condition of Step 3. in BASICMULTIFILTER would have been satisfied. This is a contradiction and completes the proof of Lemma 3.6. □

Our next lemma bounds the relative change in the weight of $S$ and $T$ if the BASICMULTIFILTER algorithm outputs two weight functions in Step 4.(b). See Appendix A for the proof.

**Lemma 3.7.** *If $T$ is $\alpha$-good and $w(S) \ge 3|S|/4$, then after Step 4.(b) of* BASICMULTIFILTER *we have that one of $w^{(1)}$ and $w^{(2)}$ will satisfy $\frac{\Delta^{(i)} w(S)}{w(S)} \le \frac{\Delta^{(i)} w(T)}{w(T)} \cdot \frac{1}{24 \lg(2/\alpha)}$ , where $\Delta^{(i)} w = w - w^{(i)}$ for $i = 1, 2$.*

Combining Lemmas 3.5 and 3.7, we obtain the following corollary.

**Corollary 3.8.** *If $T$ is $\alpha$-good and $w(S) \ge 3|S|/4$, then in each iteration of* BASICMULTIFILTER *returning new weight functions, for at least one of the new weight functions returned, we have that*

$$\frac{\Delta w(S)}{w(S)} \le \frac{\Delta w(T)}{w(T)} \frac{1}{24 \lg(2/\alpha)} \ . \tag{6}$$

The following definition facilitates the analysis in the next subsection.

**Definition 3.9 (Nice iteration).** We will call an iteration of BASICMULTIFILTER from the old weight function $w$ to the new weight function $w'$ such that (6) is satisfied a nice iteration.

### 3.3 Main Algorithm

Our main algorithm is presented in pseudocode below.

---

**Algorithm** MAINSUBROUTINE
Input: $T \subset \mathbb{R}^d$ and weight function $w$ on $T$, $0 < \alpha < 1/2$

1. Let $\Sigma_{T,w} = \text{Cov}_w[T]$ be the weighted covariance matrix.
2. Let $\lambda$ be the top eigenvalue and $v$ an associated unit eigenvector of $\Sigma_{T,w}$. Compute approximations $\lambda^*$ and $v^*$ to these satisfying $(v^*)^T \Sigma_{T,w} v^* = \lambda^*$ and $\lambda \ge \lambda^* \ge \lambda/2$.
3. Run BASICMULTIFILTER$(v^*, T, w, \alpha)$.
   (a) If it returns "YES", then return $\mu_w(T)$.
   (b) If it returns a list $\{(T, w', \alpha)\}$, then return the list containing the elements of $\{(T, w', \alpha)\}$ with $w'(T) \ge \alpha|T|/2$.

---

**Algorithm** LIST-DECODE-MEAN
Input: $T \subset \mathbb{R}^d$, $0 < \alpha < 1/2$

1. Let $L = \{(T, w^{(0)}, \alpha)\}$, where $w^{(0)}(x) = 1$ for all $x \in T$, and let $M = \emptyset$.
2. While $L \ne \emptyset$:
   (a) Get the first element $(T, w, \alpha)$ from $L$ and remove it from the list.
   (b) Run MAINSUBROUTINE$(T, w, \alpha)$.
       (a) If this routine returns a vector, then add it to $M$.
       (b) If it returns a list of $(T, w', \alpha)$, append that to $L$.
3. Output $M$ as a list of guesses for the target mean $\mu$ of $D$.

---

Our first lemma establishes that, under certain conditions, if MAINSUBROUTINE returns a hypothesis vector, this vector will be close to the target mean.

**Lemma 3.10.** *If $T$ is $\alpha$-good, $w(S) \ge 3|S|/4$, and* MAINSUBROUTINE *returns a vector $\mu_w(T)$, then we have that $\|\mu - \mu_w(T)\|_2 \le O\left(\log(1/\alpha)/\sqrt{\alpha}\right)$.*

*Proof (sketch).* We provide a sketch here, see Appendix A for the full proof.

If $\beta = w(S)/w(T)$, then for any unit vector $v$, we have that

$$\mathrm{Var}_w[v \cdot T] \geq \beta(v \cdot (\mu_w(S) - \mu_w(T)))^2 \ .$$

Because the algorithm returned a vector at this step, we have that $\mathrm{Var}_w[v \cdot T] = O(\log^2(1/\alpha))$, and by our assumptions $\beta \gg \alpha$. Together these imply that $|v \cdot (\mu_w(S) - \mu_w(T))| = O(\log(1/\alpha)/\sqrt{\alpha})$. Since this holds for all directions, $\|\mu_w(S) - \mu_w(T)\|_2 = O(\log(1/\alpha)/\sqrt{\alpha})$. Finally, since we kept a constant fraction of the mass of $S$, and since the covariance of $S$ is $O(I)$, a similar argument tells us that $\|\mu_w(S) - \mu\|_2 = O(1)$. Combining these with the triangle inequality gives the lemma. $\square$

So far, we have shown that if the algorithm LIST-DECODE-MEAN reaches a stage in which the BASICMULTIFILTER routine returns the vector $\mu_w(T)$, where the current weight function $w$ satisfies $w(S) \geq 3|S|/4$, then $\mu_w(T)$ is an accurate estimate of the target mean $\mu$. It remains to show that LIST-DECODE-MEAN will provably reach such a stage.

**Lemma 3.11.** *Assume $T$ is $\alpha$-good. Then, following a sequence of nice iterations of* BASICMULTI-FILTER *in* LIST-DECODE-MEAN *starting from the uniform weight function $w^{(0)}$, we obtain a weight function $w$ with $w(S) \geq 3|S|/4$ for which the* BASICMULTIFILTER *subroutine returns "YES".*

Due to space limitations, the proof of Lemma 3.11 is given in Appendix A. The analysis of the runtime can be found in Appendix B, where it is also shown that the output list is of size $O(1/\alpha^2)$. Finally, in Appendix C, we sketch how to efficiently post-process the output to an $O(1/\alpha)$-sized list.

## 4  Conclusions

In this paper, we study the problem of list-decodable mean estimation for bounded covariance distributions. As our main contribution, we give the first provable practical algorithm for this problem with near-optimal error guarantees. At a technical level, our work strengthens and generalizes the multi-filtering approach of [DKS18], which focused on spherical Gaussians, to apply under a bounded covariance assumption. This work is part of the broader agenda of developing fast and practical algorithms for list-decodable learning under minimal assumptions on the inliers.

The obvious open problem is to design faster provable algorithms for list-decodable mean estimation with $\tilde{O}(nd)$ as the ultimate goal. The runtime analysis of our algorithm gives a bound of $\tilde{O}(n^2 d/\alpha^2)$. We believe this can be easily improved to $\tilde{O}(n^2 d/\alpha^{1+c})$, for any constant $c > 0$. A bottleneck in our runtime analysis comes from the number of recursive subsets that our algorithm needs to run on. This is controlled by Equation (1), which postulates that $\sum w_i(T)^2 \leq |T|^2$. This condition ensures that we have no more than $O(\alpha^{-2})$ many subsets at any given time. This can be improved by replacing (1) by $w(T_1)^{1+c} + w(T_2)^{1+c} \leq w(T)^{1+c}$, for any $c > 0$. We believe this should suffice to let the remainder of our analysis go through and reduce the $\alpha$-dependence of our runtime to $O(\alpha^{-1-c})$.

The concurrent work [CMY20] gives an SDP-based algorithm whose runtime is $\tilde{O}(nd)/\mathrm{poly}(\alpha)$, i.e., near-optimal as a function of the dimension $d$, but suboptimal (by a polynomial factor) as a function of $1/\alpha$. We note that the dependence on $1/\alpha$ is equally significant in some of the key applications of list-decodable learning (e.g., in learning mixture models). Can we obtain a *truly* near-linear time algorithm?

## Broader Impact

Our work fits within a broader agenda of algorithmic high-dimensional robust statistics and aims to advance the algorithmic foundations of robust learning in the presence of a large fraction of arbitrary outliers. An important motivation for this line of work is to design provable defenses of machine learning systems against *data poisoning attacks*. This goal has become a pressing challenge in many real-world scenarios, where the data of a machine learning system can be untrusted (including, e.g., crowdsourcing).

Since the primary focus of our work is theoretical, we do not expect our results to have immediate societal impact. Nonetheless, we believe that our algorithm is practical and that our findings provide interesting insights that could be useful in the design of practically relevant estimators in highly noisy environments.

## Acknowledgments and Disclosure of Funding

We thank Alistair Stewart for his contributions in the early stages of this work.

Ilias Diakonikolas is supported by NSF Award CCF-1652862 (CAREER) and a Sloan Research Fellowship. Daniel M. Kane is supported by NSF Award CCF-1553288 (CAREER) and a Sloan Research Fellowship.

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
