[Supplementary Material]

# Supplementary Material

## A  Proofs Omitted from Section 3

### A.1  Proof of Lemma 3.5

Since $S$ is representative and $w(S) \geq 3|S|/4 \geq |S|/2$, we see that

$$\text{Var}_w[v \cdot S] \leq \frac{1}{w(S)} \sum_{x \in S} (v \cdot x - v \cdot \mu(S))^2 \leq \frac{2}{|S|} \sum_{x \in S} (v \cdot x - v \cdot \mu(S))^2 = 2\,\text{Var}[v \cdot S] \leq 2 \ .$$

If $v \cdot \mu_w(S) \notin [a - 2, b + 2]$, then

$$\text{Var}_w[v \cdot S] \geq \frac{1}{w(S)} \sum_{\substack{x \in S \\ v \cdot x \in [a,b]}} w(x)(v \cdot x - v \cdot \mu_w(S))^2 > \frac{1}{w(S)} \sum_{\substack{x \in S \\ v \cdot x \in [a,b]}} 4w(x) \geq \frac{4}{2} = 2 \ ,$$

since $w(\{x \in S : v \cdot x \in [a, b]\}) \geq w(S) - \alpha w(T)/4 \geq 3|S|/4 - \alpha|T|/4 \geq |S|/2 \geq w(S)/2$ (since $\alpha w(T)/4 \leq \alpha|T|/4 \leq |S|/4$ because $T$ is $\alpha$-good), a contradiction. Hence, we have that $v \cdot \mu_w(S) \in [a - O(1), b + O(1)]$.

We note that if

$$\sum_{x \in S} w(x)f(x) \geq \frac{w(S)}{24w(T)\lg(2/\alpha)} \sum_{x \in T} w(x)f(x) \ ,$$

then

$$\sum_{x \in T} w(x)f(x) \leq \frac{24w(T)\lg(2/\alpha)}{w(S)} \sum_{x \in S} w(x)f(x) \ ,$$

where

$$\sum_{x \in S} w(x)f(x) \leq \sum_{x \in S} w(x)\big((v \cdot x - v \cdot \mu_w(S)) + O(1)\big)^2 = O\big(w(S)\,\text{Var}_w[v \cdot S] + w(S)\big) \leq O(w(S)) \ .$$

Thus, we can write

$$\begin{aligned}
\text{Var}_w[v \cdot T] &\leq \frac{1}{w(T)} \sum_{x \in T} w(x)\big(v \cdot x - \mathbb{E}_w[v \cdot T \cap 2I]\big)^2 \\
&\leq \frac{1}{w(\{x \in T : v \cdot x \in 2I\})} \sum_{\substack{x \in T \\ v \cdot x \in 2I}} w(x)\big(v \cdot x - \mathbb{E}_w[v \cdot T \cap 2I]\big)^2 \\
&\quad + \frac{1}{w(T)} \sum_{\substack{x \in T \\ v \cdot x \notin 2I}} w(x)\big(v \cdot x - \mathbb{E}_w[v \cdot T \cap 2I]\big)^2 \\
&\leq \text{Var}_w[v \cdot T \cap 2I] + \frac{1}{w(T)} \sum_{\substack{x \in T \\ v \cdot x \notin 2I}} w(x)O(f(x)) \\
&\leq C\log(2/\alpha)^2 + O\Big(\frac{1}{w(T)} \sum_{x \in T} w(x)f(x)\Big) \\
&\leq C\log(2/\alpha)^2 + O\Big(\frac{24\lg(2/\alpha)}{w(S)} \sum_{x \in S} w(x)f(x)\Big) \\
&\leq C\log(2/\alpha)^2 + O\Big(\frac{24\lg(2/\alpha)O(w(S))}{w(S)}\Big) \\
&= C\log(2/\alpha)^2 + O(\log(2/\alpha)) \leq C\log(2/\alpha)^2 + O(\log(2/\alpha)^2) \ ,
\end{aligned}$$

which is a contradiction, as this is the condition of Step 3.(a).

Hence, we have shown that

$$\sum_{x \in S} w(x)f(x) \leq \frac{w(S)}{24w(T)\lg(2/\alpha)} \sum_{x \in T} w(x)f(x) \ ,$$

and thus

$$w(x) - w_{\text{new}}(x) = \frac{f(x)}{\max_{x \in T} f(x)} w(x) \ ,$$

implies that

$$\frac{\Delta w(S)}{\Delta w(T)} = \frac{\sum_{x \in S}(w(x) - w_{\text{new}}(x))}{\sum_{x \in T}(w(x) - w_{\text{new}}(x))} = \frac{\sum_{x \in S} w(x)f(x)}{\sum_{x \in T} w(x)f(x)} \leq \frac{w(S)}{24w(T)\lg(2/\alpha)} \ ,$$

which completes the proof of Lemma 3.5.

## A.2 Proof of Lemma 3.7

Recall from the proof of Lemma 3.5 that $\text{Var}_w[v \cdot S] \leq 2$. So choosing $i \in \{1, 2\}$ such that

$$v \cdot T \cap (v \cdot \mu_w(S) - R, v \cdot \mu_w(S) + R) \subseteq v \cdot T_i \ ,$$

where the $T_i$ are as in Step 4.(a), we get that

$$\frac{\Delta^{(i)}w(S)}{w(S)} = \underset{x \in S}{w\text{-}\Pr}[x \notin T_i] \leq \underset{x \in S}{w\text{-}\Pr}[|v \cdot x - v \cdot \mu_w(S)| > R] \leq \frac{\text{Var}_w[v \cdot S]}{R^2} \leq \frac{2}{R^2} \ ,$$

and thus $\Delta^{(i)}w(S) \leq 2w(S)/R^2$ for one of $i = 1, 2$. For this $i$, we also have

$$\Delta^{(i)}w(T) = w(T)\big(1 - w^{(i)}(T)/w(T)\big) \geq w(T)48\lg(2/\alpha)/R^2 \ ,$$

so

$$\frac{\Delta^{(i)}w(S)}{\Delta^{(i)}w(T)} \leq \frac{w(S)}{w(T)} \cdot \frac{1}{24\lg(2/\alpha)} \ ,$$

which is equivalent to the claim Lemma 3.7.

## A.3 Proof of Lemma 3.10

We start with the following claim:

**Claim A.1.** *Any unit vector $v$ has $\mathbb{E}_w[(v \cdot (T - \mu_w(T)))^2] \leq 2\lambda^*$ and $\mathbb{E}_w[(v^* \cdot (T - \mu_w(T)))^2] = \lambda^*$. If MAINSUBROUTINE$(T, w, \alpha)$ returns a vector, then $\lambda^* = O(\log(1/\alpha)^2)$.*

*Proof.* We note that $\mathbb{E}_w[(v \cdot (T - \mu_w(T)))^2] = \text{Var}_w[v \cdot T] = v^T \Sigma_{T,w} v \leq \lambda \|v\|_2^2 = \lambda \leq 2\lambda^*$. In the case that $v = v^*$, we see that $(v^*)^T \Sigma_{T,w} v^* = \lambda^*$, and when MAINSUBROUTINE returns a vector, then $\text{Var}_w[v^* \cdot T] = O(\log(1/\alpha)^2)$ by construction. $\square$

We will give a slightly different proof of Lemma 3.10 than the one sketched in Section 3.

*Proof (of Lemma 3.10).* If $T$ is $\alpha$-good, we have that $w(S) \geq 3|S|/4 \geq |S|/2 \geq \alpha|T|/2 \geq \alpha w(T)/2$. Therefore,

$$\mathbb{E}_w[(v \cdot (S - \mu_w(T)))^2] = \frac{1}{w(S)} \sum_{x \in S} w(x)(v \cdot (x - \mu_w(T)))^2 \leq \frac{2}{\alpha w(T)} \sum_{x \in T} w(x)(v \cdot (x - \mu_w(T)))^2$$

$$= \frac{2\mathbb{E}_w[(v \cdot (T - \mu_w(T)))^2]}{\alpha} \leq \frac{4\lambda^*}{\alpha} \ ,$$

i.e.,

$$\mathbb{E}_w[(v \cdot (S - \mu_w(T)))^2] \leq 4\lambda^*/\alpha \ . \tag{7}$$

Note that $\mathbb{E}[v \cdot (S - \mu_w(T))] = v \cdot (\mu(S) - \mu_w(T))$ and $\text{Var}[v \cdot (S - \mu_w(T))] = \text{Var}[v \cdot S] \leq 1$. So by Cantelli's inequality

$$\Pr[v \cdot (S - \mu_w(T)) \geq v \cdot (\mu(S) - \mu_w(T)) - 1] \geq 1 - \frac{\text{Var}[v \cdot (S - \mu_w(T))]}{\text{Var}[v \cdot (S - \mu_w(T))] + 1} \geq 1 - 1/2 = 1/2,$$

since $x/(x + 1)$ is increasing on $(0, \infty)$ so

$$\frac{\text{Var}[v \cdot (S - \mu_w(T))]}{\text{Var}[v \cdot (S - \mu_w(T))] + 1} \leq 1/2 \ .$$

Now $w(S) \geq 3|S|/4$, so

$$w\text{-Pr}[v \cdot (S - \mu_w(T)) \geq v \cdot (\mu(S) - \mu_w(T)) - 1] \geq 1/4.$$

By Markov's inequality applied to (7), we get

$$w\text{-Pr}[v \cdot (S - \mu_w(T)) \geq \sqrt{16\lambda^*/\alpha}] \leq \frac{\mathbb{E}_w[(v \cdot (S - \mu_w(T)))^2]}{(\sqrt{16\lambda^*/\alpha})^2} \leq \frac{4\lambda^*/\alpha}{16\lambda^*/\alpha} = \frac{1}{4},$$

and so

$$v \cdot (\mu(S) - \mu_w(T)) - 1 \leq \sqrt{16\lambda^*/\alpha}.$$

Thus, if MainSubroutine returns a vector, we have by Claim A.1 that

$$v \cdot (\mu(S) - \mu_w(T)) \leq 1 + \sqrt{16\lambda^*/\alpha} \leq O\left(\log(2/\alpha)/\sqrt{\alpha}\right)$$

for all unit vectors $v$. Hence, we obtain

$$\|\mu - \mu_w(T)\|_2 \leq \|\mu - \mu(S)\|_2 + \|\mu(S) - \mu_w(T)\|_2 \leq 1 + O\left(\log(2/\alpha)/\sqrt{\alpha}\right) = O\left(\log(1/\alpha)/\sqrt{\alpha}\right).$$

$\square$

## A.4 Proof of Lemma 3.11

Let $w^{(0)}$ be the weight function on $T$ given by $w^{(0)}(x) = 1$ for all $x \in T$ so that $w^{(0)}(T) = |T|$, and let $A_k = \left\{ w \colon T \to [0,1] \mid |T|/2^k < w(T) \leq |T|/2^{k-1} \right\}$ for $k \geq 1$.

Clearly, $w^{(0)}(S) = |S| \geq 3|S|/4$, so by Corollary 3.8 the BasicMultifilter there is a nice first iteration.

Suppose that we have a sequence

$$w^{(i)} \xrightarrow{\Delta^{(i+1)}w} w^{(i+1)} \xrightarrow{\Delta^{(i+2)}w} \cdots \xrightarrow{\Delta^{(j)}w} w^{(j)}$$

of nice iterations, where $w^{(i)}, w^{(i+1)}, \dots, w^{(j)} \in A_k$ for some fixed $k$. Then, by Corollary 3.8 and the definition of $A_k$, we get that

$$\sum_{m=i}^{j-1} \Delta^{(m+1)}w(S) \leq \sum_{m=i}^{j-1} \frac{w^{(m)}(S)}{w^{(m)}(T)} \cdot \frac{1}{24\lg(2/\alpha)} \Delta^{(m+1)}w(T) \leq \frac{|S|}{|T|} \frac{2^k}{24\lg(2/\alpha)} \sum_{m=i}^{j-1} \Delta^{(m+1)}w(T)$$

$$\leq \frac{|S|}{|T|} \frac{2^k}{24\lg(1/\alpha)} \frac{|T|}{2^k} = \frac{|S|}{24\lg(2/\alpha)}.$$

Now suppose on the other hand that we have a nice iteration

$$w^{(j)} \xrightarrow{\Delta^{(j+1)}w} w^{(j+1)},$$

where $w^{(j)} \in A_k$ and $w^{(j+1)} \in A_{k+r}$ for some $r \geq 1$. Then, by Corollary 3.8 and the definition of $A_k$, we get that

$$\Delta^{(j+1)}w(S) \leq \frac{w^{(j)}(S)}{w^{(j)}(T)} \cdot \frac{1}{24\lg(2/\alpha)} \Delta^{(j+1)}w(T) \leq \frac{|S|}{|T|} \frac{2^k}{24\lg(2/\alpha)} \left(\frac{1}{2^{k-1}} - \frac{1}{2^{k+r}}\right)|T| \leq 2\frac{|S|}{24\lg(2/\alpha)}.$$

Now suppose that we have gotten to iteration $m$ via a sequence of nice iterations

$$w^{(0)} \xrightarrow{\Delta^{(1)}w} w^{(1)} \xrightarrow{\Delta^{(2)}w} \cdots \xrightarrow{\Delta^{(m)}w} w^{(m)},$$

where $w^{(i)}(S) \geq 3|S|/4$ for all $i \leq m-1$. We want to show that $w^{(m)}(S) \geq 3|S|/4$. First, we note that $w^{(m-1)} \in A_k$ for some $k \leq \lg(2/\alpha)$, since $1/2^{\lg(2/\alpha)} = 1/(2/\alpha) = \alpha/2$ and $w^{(m-1)}(T) \geq 3|S|/4 > |S|/2 \geq \alpha|T|/2$. So we get that we have at most $k+1$ iterations $A_\ell \to A_{\ell+r}$ ($r \geq 1$) (the worst case being $A_0 \to A_1 \to \cdots \to A_k \to A_{k+r}$), and thus by the above

$$\sum_{s=0}^{m-1} \Delta^{(s+1)}w(S) \leq \frac{3(k+1)|S|}{24\lg(2/\alpha)} \leq \frac{(\lg(2/\alpha)+1)|S|}{8\lg(2/\alpha)} \leq \frac{2\lg(2/\alpha)|S|}{8\lg(2/\alpha)} \leq \frac{|S|}{4}.$$

Therefore, we have that

$$w^{(m)}(S) = |S| - \sum_{s=0}^{m-1} \Delta^{(s+1)} w(S) \geq 3|S|/4 \ .$$

Hence, by Corollary 3.8, we get another nice iteration.

By induction, every sequence of nice iterations

$$w^{(0)} \xrightarrow{\Delta^{(1)} w} w^{(1)} \xrightarrow{\Delta^{(2)} w} \cdots \xrightarrow{\Delta^{(m)} w} w^{(m)}$$

satisfies $w^{(i)}(S) \geq 3|S|/4$ for all $i$, and there is a sequence of purely nice iterations.

Hence, every weight function $w$ in a sequence of nice iterations starting from $w^{(0)}$ satisfies $w(T) \geq w(S) \geq 3|S|/4 > |S|/2 \geq \alpha|T|/2$ (since $T$ is $\alpha$-good), and thus we cannot get to an iteration where $w(T) < \alpha|T|/2$. This implies that we have to exit the algorithm by getting "YES" in the BASICMULTIFILTER (since every branch of the algorithm terminates). This completes the proof.

## B  Runtime Analysis

In this section, we provide a detailed runtime analysis of our main algorithm. We start with the following simple lemma.

**Lemma B.1.** BASICMULTIFILTER *has worst-case runtime* $\tilde{O}(|T|d)$.

*Proof.* The operations in BASICMULTIFILTER can be implemented efficiently with an appropriate preprocessing. In particular, computing $v \cdot x$ for each $x \in T$ can be done in $O(d|T|)$ time, and then sorting these values can be done in $O(|T| \log(|T|))$ time. Then in $O(|T|)$ time, using a linear scan, we can compute and store $w(\{x \in T : v \cdot x \leq v \cdot y\})$ for all $y \in T$. Computing $a$ and $b$ in Step 2. can be done in $O(\log(|T|))$ time by binary search. The variance of $v \cdot T$ over $T$ conditioned on $I$ can be done in linear time in the usual way. The computation of $w_{new}$ in Step 3.(b) is easily done in linear time.

The most challenging part of the algorithm is Step 4.. Assuming that there is a solution, we let $a$ be the smallest value in $v \cdot T_1$ and $b$ the largest value in $v \cdot T_2$. We will have our algorithm guess which of $w(T_1)$ or $w(T_2)$ is larger, thus determining which term achieves the minimum in Equation (2). If $w(T_1)$ is larger, we additionally guess the value of $a$ and if $w(T_2)$ is larger, we guess $b$. We note that there are $O(|T|)$ many possible outcomes for these guesses and that upon making them we can determine the value of $\min(1 - w(T_1)/w(T), 1 - w(T_2)/w(T))$. This lets us determine the largest possible value of $R$ consistent with condition (2). This in turn lets us determine the smallest possible value of $b$ (if we guessed $a$), or the largest possible value of $a$ (if we guessed $b$) consistent with these guesses, and condition (2) by using binary search to find the largest/smallest element of $v \cdot T$ so that $b - a \geq 2R$ and so that the $w(T_i)$ chosen to attain the minimum actually does. Note then that if any choices of $t$ and $R$ consistent with our guess and with condition (2) are also consistent with condition (1), this extreme choice will be. Therefore, it suffices for each of these $O(|T|)$ possible guesses to spend $O(\log(|T|))$ time to find this extreme value, and then spend $O(1)$ time to verify whether or not condition (1) and (2) hold. Once we find some choice for which they do, we can return that one. The total runtime for this step is at most $O(|T| \log(|T|))$. $\qquad \square$

**Lemma B.2.** MAINSUBROUTINE *has worst case runtime* $\tilde{O}(|T|d)$.

*Proof.* We note that Steps 1 and 2 can be implemented in time $\tilde{O}(|T|d)$ by standard methods. In particular, we do not need to explicitly compute the weighted empirical covariance. We can instead use power-iteration to find an approximately largest eigenvalue-eigenvector pair in $\tilde{O}(|T|d)$ time. Even though this computation is randomized, we can ignore the error probability for the following reason: By standard linear-algebraic tools (see, e.g., Fact 5.1.1 of [Li18]), this computation takes time $\tilde{O}(|T|d \log(1/\delta))$, where $\delta$ is the error probability. Since we only use this subroutine $|T|$ many times, we can take $\delta \ll 1/|T|$ and use a union bound. This incurs at most a logarithmic overhead in the running time.

By Lemma B.1, Step 3 can be completed in $\tilde{O}(|T|d)$ time, which completes the proof. $\qquad \square$

We are now ready to prove the main theorem of this section:

**Theorem B.3.** LIST-DECODE *has worst-case runtime* $\tilde{O}(|T|^2 d/\alpha^2)$ *and the output list $M$ has size at most* $4/\alpha^2$.

*Proof.* We note that the multi-filter algorithm gives us the structure of a tree, wherein, by Equation (1), we get that

$$|T|^2 = w^{(0)}(T)^2 \geq \sum_{\text{all leaves } w} w(T)^2 \geq \sum_{\text{all leaves } w} (\alpha|T|/2)^2 ,$$

and thus

$$4/\alpha^2 \geq \sum_{\text{all leaves } w} 1 = \#\text{of leaves}.$$

So we have at most $4/\alpha^2$ leaves and thus at most $4/\alpha^2$ elements in the list $M$. Also, by the above, we never have more than $O(\alpha^{-2})$ branches at a given depth in the tree.

The bottleneck of the algorithm is clearly in Step 2. Each call to MAINSUBROUTINE can be completed with runtime $\tilde{O}(|T|d)$ by Lemma B.2, so we just need to consider how many rounds of Step 2 we can have in the worst case. We note that each iteration of BASICMULTIFILTER sets at least one weight to $0$ (in every branch), so the tree has depth at most $|T|$, and therefore we run Step 2 at most $O(|T|\alpha^{-2})$ times, since we never have more than $O(\alpha^{-2})$ branches. Hence, the runtime is $\tilde{O}(|T|^2 d\alpha^{-2})$. $\quad\square$

## C Efficient List Size Reduction

Here we give a simple and efficient method to reduce the list of hypotheses to one of size $O(1/\alpha)$.

**Theorem C.1.** *There exists an algorithm that given the output of Theorem 3.3 runs in time $O(d/\alpha^3)$ and returns a list of $O(1/\alpha)$ hypotheses with the guarantee that at least one of the hypotheses are within $O(\log(1/\alpha)/\sqrt{\alpha})$ of $\mu$, assuming that one of the hypotheses of the original algorithm was.*

The algorithm here is quite simple. We set $C > 0$ to be a sufficiently large universal constant and find a maximal subset of our hypotheses that are pairwise separated by at least $C\log(1/\alpha)/\sqrt{\alpha}$. This can be done by starting with an empty set $H$ of hypotheses and for each hypothesis in the output of Theorem 3.3 comparing it to each of the hypotheses currently in $H$ and adding it if it is not too close to any of them. It is clear that the runtime of such an algorithm is at most $O(d|H|/\alpha^2)$. It is also clear that if our original set of hypotheses contained a $\mu_0$, then $H$ will contain a $\tilde{\mu}$ with $\|\tilde{\mu} - \mu_0\|_2 < C\log(1/\alpha)/\sqrt{\alpha}$. Therefore, if our original set contained a $\mu_0$ with $\|\mu_0 - \mu\|_2 = O(\log(1/\alpha)/\sqrt{\alpha})$, then by the triangle inequality, $H$ will contain a $\tilde{\mu}$ with $\|\tilde{\mu} - \mu\|_2 = O(\log(1/\alpha)/\sqrt{\alpha})$.

All we have left to prove is that $|H| = O(1/\alpha)$. For this we note that for each hypothesis $\mu_i$ that Theorem 3.3 returns, there is an associated weight function $w_i$ on $T$ so that

- $w_i(T) \geq \alpha|T|/2$,
- $\mathrm{Cov}_{w_i}[T] \prec O(\log^2(1/\alpha))I$.

It turns out that this is enough to show that $|H| = O(1/\alpha)$. This argument has become fairly standard in the robust list-decoding literature, but unfortunately, we cannot find an existing theorem statement that applies to exactly our case. The techniques in the proof of Claim 5.2 of [DKS18] are very similar. We state a general theorem here that not only covers our case, but should cover more general settings:

**Lemma C.2.** *Let $T$ be a subset of $\mathbb{R}^d$ and $\alpha, \sigma > 0$ be real numbers. Let $H$ be another subset of $\mathbb{R}^d$ such that for each $u \in H$ there is a weight function $w_u$ on $T$ with $w_u(T) \geq \alpha|T|$ and such that for any unit vector $v \in \mathbb{R}^d$, $w\text{-Pr}_{x \in T}[|v \cdot (x - u)| > \sigma] < \alpha/10$. Assume furthermore that for any $u, u' \in H$, we have $\|u - u'\|_2 > 2\sigma$. Then we have that $|H| \leq 2/\alpha$.*

Applying this lemma to our set $H$ with the weight functions $w_i$ mentioned above and $\sigma$ a sufficiently large multiple of $\log(1/\alpha)/\sqrt{\alpha}$ yields Theorem C.1.

Before we begin, we will introduce the notation $\bigcup_{i \in I} w_i$ and $\bigcap_{i \in I} w_i$ for the weight functions given by

$$\left(\bigcup_{i \in I} w_i\right)(x) = \max_{i \in I} w_i(x),$$

$$\left(\bigcap_{i \in I} w_i\right)(x) = \min_{i \in I} w_i(x)$$

for any finite index set $I$ and any $x \in T$.

*Proof.* We proceed by contradiction. Assume that the above hypotheses hold and that $|H| > 2/\alpha$. We note that $(\bigcup_{u \in H} w_u)(T) \le |T|$. However, the sum of the individual terms is much larger than this since $\sum_{u \in H} w_u(T) \ge \sum_{u \in H} \alpha|T| \ge 2|T|$ because $|H| \ge 2/\alpha$. By restricting $H$ to a subset if necessary, we can guarantee that $|H| = \lceil 2/\alpha \rceil$, which will still ensure that $\sum_{u \in H} w_u(T) \ge 2|T|$. Next, as we will show, the pairwise intersections of the $w_u$ are small. This will give a contradiction.

To start with, note that given $u, u' \in H$ we would like to show that $(w_u \cap w_{u'})(T)$ is small. For this, we let $v$ be the unit vector in the direction of $u - u'$. By assumption, $v \cdot (u - u') = \|u - u'\|_2 > 2\sigma$. Therefore, by the triangle inequality, for every $x \in T$ it will either be the case that $|v \cdot (x - u)| > \sigma$ or $|v \cdot (x - u')| > \sigma$. However, if we call these sets $D_u$ and $D_{u'}$, we have that $w_u(D_u) \le \alpha/10\, w_u(T)$ and $w_{u'}(D_{u'}) \le w_{u'}(T)$. Therefore, we have that $(w_u \cap w_{u'})(T) \le \alpha(w_u(T) + w_{u'}(T))/10$.

Given this, we wish to make use of approximate inclusion-exclusion. In particular, given some ordering over the points in $H$, we note that for any $x \in T$ we have that

$$1 \ge \max_{u \in H} w_u(x) \ge \sum_{u \in H} w_u(x) - \sum_{u, u' \in H, u < u'} \min(w_u(x), w_{u'}(x)).$$

Summing over $x \in T$, we find that

$$
\begin{aligned}
|T| &\ge \Big(\bigcup_{u \in H} w_u\Big)(T) \\
&\ge \sum_{u \in H} w_u(T) - \sum_{u, u' \in H, u < u'} (w_u \cap w_{u'})(T) \\
&\ge \sum_{u \in H} w_u(T) - \sum_{u, u' \in H, u < u'} (\alpha/10)\big(w_u(T) + w_{u'}(T)\big) \\
&= \Big(\sum_{u \in H} w_u(T)\Big)(1 - (\alpha/10)(|H| - 1)) \\
&\ge \Big(\sum_{u \in H} w_u(T)\Big)(1 - (\alpha/10)(2/\alpha)) \\
&\ge 2|T|(8/10) \\
&> |T|,
\end{aligned}
$$

where we use that $|H| - 1 \le 2/\alpha$ and $\sum_{u \in H} w_u(T) \ge 2|T|$ (after we restricted $H$ above), yielding a contradiction. Hence $|H| \le 2/\alpha$.

This completes our proof. $\qquad\square$