[Reviews · NeurIPS 2020]

Review 1

Summary and Contributions: This paper studies *list-decodable mean estimation with bounded covariance*, a basic problem in outlier-robust statistics. Given a collection of vectors in $R^d$ of which an $\alpha$ fraction for some small constant $\alpha < 1/2$ are sampled from a probability distribution of bounded covariance and the rest may be chosen adversarially, the goal is to estimate the mean of that distribution. Since this is not information-theoretically possible, the algorithm is allowed to output a list of $1/\alpha$ "candidate" means, such that the true mean is close to one on the list. This problem generalizes learning mixtures of bounded-covariance distributions in the outlier-robust setting. While polynomial-time algorithms with optimal or nearly-optimal error guarantees for this problem have been known for a few years (e.g. by Steinhardt-Charikar,Valiant), no known algorithm so far comes with a practical running time. The main contribution of this paper is the first algorithm for this problem with nearly-optimal error guarantees and a reasonably practical running time -- $n^2 d / \alpha^2$, where $n$ is the number of samples, $d$ is the dimension, and $\alpha$ is the fraction of "good" samples. The algorithm is based on a variant of the spectral iterative downweighting schemes which have become popular in the robust statistics literature, usually going by the name "filter". The "multifilter" version here, adapted to the case of $\alpha < 1/2$ (where one must identify a list of means instead of just a single mean vector), appears first in work by Diakonikolas-Kane-Stewart of 2018, where it is analyzed only for the case of Gaussians, and the fast running time here is not obtained. The present work modifies the multifilter to tolerate the much weaker assumption of bounded covariance, and improves the running time analysis. UPDATE: Thanks to the authors for the detailed response. I have re-read the relevant lines of the submission; I understand better now that there is some significant technical difficulty in extending the multifilter to the bounded covariance case. (That said, I don't find the "technical overview" to be very accessible...it is kind of a wall of text.) Anyway, I remain happy to support accepting the paper.

Strengths: The paper studies a basic problem in robust statistics which should be of interest to the NeurIPS audience. The error guarantees it achieves are optimal up to logarithmic factors. The algorithm is the first to maintain strong provable guarantees while remaining reasonably practical.

Weaknesses: In light of the DKS '18 paper as well as the existence of fast, practical algorithms for robust mean estimation in the $\alpha > 1/2$ case, it is not clear how much technical novelty there is here. However, I think the result on its own is probably interesting enough to merit acceptance, even if the techniques are not too surprising. Since the algorithm is supposed to be "practical", it would have been nice to see some numerical experiments -- it's quite disappointing that there are none here.

Correctness: I did not attempt to verify the arguments line by line, but judging by the algorithm's description it seems quite likely to be correct; all the key ingredients seem to be present.

Clarity: The paper is reasonably well written.

Relation to Prior Work: The discussion is adequate. It would be nice if the authors updated the manuscript to include a comparison to the recent work on the same subject to appear in FOCS 2020.

Reproducibility: Yes

Additional Feedback:


Review 2

Summary and Contributions: This paper studies the problem of robust mean estimation of bounded-covariance distributions in the list-decodable setting where 1 - alpha fraction of the points are corrupted for alpha < 1/2. Prior work of Charikar-Steinhardt-Valiant that introduced this problem gave a sample-optimal algorithm achieving the information-theoretically optimal error rate, i.e. outputting an O(1/alpha)-sized list containing an O~(1/sqrt(alpha))-close estimate, but suffering an impractical polynomial runtime. The present paper gives an algorithm that gets the same statistical guarantees while running in time n^2*d/poly(alpha) where n = O(d/alpha) is the sample size and d is the dimension. The algorithm is a more sophisticated variant of the multifilter originally introduced by Diakonikolas-Kane-Stewart '18 for the version of this problem where the distribution is isotropic Gaussian. At a high level, like the iterative filtering framework for robust mean estimation in the large-alpha regime, they repeatedly project in the direction of largest variance and run some kind of outlier removal along that direction. The main difference in the list-decodable setting is that one cannot afford to always make a single decision of which points to remove in a given iteration, as there could be many different true means that are equally viable. Instead, the idea is that unless the projected points are very tightly concentrated (in which case the empirical mean of the remaining points is a good estimator), there should exist two overlapping intervals such that removing the points outside _one_ of the intervals will throw out mostly corrupted points, and then one can repeat. The heavy tails of projections of the underlying distribution make designing and analyzing the multifilter more difficult in the present work than in the Gaussian case. Indeed, one cannot afford to do _hard_ outlier removal, so, similar to the more recent papers in this area, they instead maintain weights over the data and iteratively scale them down. Rather than try to show that one of the intervals output at each step is such that throwing out the complement removes at most poly(alpha) good points, they show that significantly more bad mass is thrown out than good mass at each step. UPDATE: Thanks for the explanation of how this work compares with CMY. I continue to be in favor of accepting this paper.

Strengths: The mutlifilter of DKS18 was a very nice idea, and it's good to see it being extended to give new applications in this literature. Given the relevance of list-decodable mean estimation to clustering mixture models, it is also important to develop practical algorithms for this basic question, and this work takes an important step towards this goal. The length of this submission belies the incredible sophistication of the analysis, and it's an impressive technical feat to be able to extend the analysis from DKS18 to handle the heavy tailed case.

Weaknesses: The main areas for improvement are 1) there is an extra log(1/alpha) factor in the error guarantee, and 2) the algorithm, like the original instantiations of the iterative filtering framework, needs to do O(n) passes over the dataset, leading to an n^2 * d runtime. Very recent concurrent work of Cherapnamjeri-Mohanty-Yau in FOCS '20 shows that both of these can be avoided (using a different algorithm), at the cost of a super-cubic dependence on 1/alpha in the runtime.

Correctness: I have verified that the proofs are correct.

Clarity: Overall, the paper is quite well written. A minor complaint was that the proof of Lemma 3.6 was a bit hard for me to digest. The intuitive explanations in some of the informal proofs in the main submission were quite helpful, and it would be nice to include a similar explanation of what's going on there.

Relation to Prior Work: The paper clearly situates itself in the burgeoning robust statistics literature. The only reference that the authors should also mention is the abovementioned recent paper of Mohanty-Cherapnamjeri-Yau which achieves qualitatively similar results.

Reproducibility: Yes

Additional Feedback: Question: Not asking to run any experiments, but are there any bottlenecks for getting a working implementation of this algorithm for reasonably large alpha < 1/2? Typos: line 117: "withe" -> "with"


Review 3

Summary and Contributions: The paper deals with the problem of robustly estimating the mean of a multidimensional distribution, an extensively studied problem in the recent literature. This problem is of fundamental importance in practical applications of machine learning. The (practical) computational complexity if currently one of the main deadlock despite promising previous results based on SoS and Semi-definite programming. The present paper provides an iterative spectral approach which seems scalable and useful for practitioners.

Strengths: The paper's contribution is worthwhile, in my opinion, given the poor provable scalability of previous algorithms in the field. Rigorous proofs are provided for the claimed results. The proposed approach seems novel to me.

Weaknesses: I would have appreciate a wider account of previous work in algorithmic robust estimation.

Correctness: The proofs I have reviewed are correct.

Clarity: The paper is well written.

Relation to Prior Work: Relationship with prior work is clearly stated. I would have appreciated comparisons with other techniques such as Median of Means type of approaches. UPDATE: Thanks for the response concerning this point.

Reproducibility: Yes

Additional Feedback:

[Author Response · NeurIPS 2020]

We thank the reviewers for their careful consideration of our paper and their positive feedback. Below we address
individual comments/questions by the reviewers.

**Reviewer 1:** We thank the reviewer for their positive feedback.

*Technical novelty:* We would like to point out that "multi-filtering" is an algorithmic framework and not a specific
algorithm. As the reviewer notes, this framework was first introduced in a STOC'18 paper by Diakonikolas, Kane, and
Stewart. The [DKS18] paper obtained list-decodable mean estimation algorithms for *identity covariance Gaussian*
*distributions*. We emphasize that enhancing this framework to obtain our efficient algorithm (with near-optimal error
guarantees) for the broad family of *bounded (and unknown) covariance distributions* requires overcoming a number of
obstacles. To do so, we develop new technical tools that were not present in [DKS18] or in the heavy-tailed robust
mean estimation algorithms for $\alpha > 1/2$. Please see lines 105-160 of our submission for an overview.

We termed our algorithm "potentially practical", given that it is iterative (with each iteration being fast, running in
near-linear time) and based on prior experience regarding the experimental performance of filtering algorithms. We
agree with the reviewer that an experimental evaluation of our algorithm would be interesting.

*Related work:* We will make sure to cite and compare with the contemporaneous work by Cherapanamjeri, Mohanty,
and Yau in the final version of our paper. We would like to point out that this work first appeared on the arXiv around
a week before the NeurIPS deadline. Briefly, we note that this concurrent work uses very different techniques and
achieves runtime $\tilde{O}(nd)/\operatorname{poly}(\alpha)$, for some unspecified degree polynomial in $1/\alpha$ (from our reading, the degree of the
$\operatorname{poly}(\alpha)$ dependence seems to be at least six). This runtime is better than that of our algorithm as a function of $n$, but
worse as a function of $1/\alpha$. We note that the runtime dependence on $1/\alpha$ is equally significant in some key applications
of list-decodable learning (e.g., in learning mixture models with many components).

**Reviewer 2:** We thank the reviewer for their positive feedback.

Regarding comparison to contemporaneous related work: Please see the *Related Work* paragraph in the response to
Reviewer 1.

Regarding implementability of our algorithm: Our algorithm is simple to implement and we believe it can scale in high
dimensions, even for small values of the parameter $\alpha < 1/2$. That said, an experimental evaluation is left for future
work. The contribution of our paper is theoretical and boils-down to developing the first algorithm for this fundamental
learning problem that avoids the ellipsoid method and has a low-degree polynomial runtime.

**Reviewer 3:** We thank the reviewer for their positive feedback.

Due to space limitations, in Section 1.1 we provide the necessary background. In particular, we summarize the prior
work in algorithmic high-dimensional robust statistics that is most closely related to the results of our paper. We will be
happy to add more context in the final version.

The median-of-means based techniques are typically used in a related, but different, context. The prototypical example
is when we have i.i.d. samples from a heavy-tailed (bounded covariance) distribution and the goal is to obtain a
*high-confidence* estimator of its mean. This notion of robustness is different than the one considered in our paper where
the majority of the input dataset consists of adversarial outliers. We will clarify this connection in our final version.

[Meta-Review · NeurIPS 2020]

Congratulations -- the reviewers all enjoyed your paper and appreciated the contribution.